# NF-κB and TNF Affect the Astrocytic Differentiation from Neural Stem Cells

**DOI:** 10.3390/cells10040840

**Published:** 2021-04-08

**Authors:** Cindy Birck, Aurélien Ginolhac, Maria Angeliki S. Pavlou, Alessandro Michelucci, Paul Heuschling, Luc Grandbarbe

**Affiliations:** 1Department of Life Sciences and Medicine, Faculty of Science, Technology and Communication, University of Luxembourg, L-1511 Luxembourg, Luxembourg; cindy.birck@gmail.com (C.B.); aurelien.ginolhac@uni.lu (A.G.); maria.pavlou@uni.lu (M.A.S.P.); paul.heuschling@uni.lu (P.H.); 2NORLUX Neuro-Oncology Laboratory, Department of Oncology, Luxembourg Institute of Health, L-1526 Luxembourg, Luxembourg; Alessandro.Michelucci@lih.lu; 3Neuro-Immunology Group, Department of Oncology, Luxembourg Institute of Health, L-1526 Luxembourg, Luxembourg; 4Luxembourg Centre for Systems Biomedicine, University of Luxembourg, L-4362 Esch-sur-Alzette, Luxembourg

**Keywords:** astrocyte, neural progenitor cell, NF-κB, tumor necrosis factor, differentiation, apoptosis

## Abstract

The NF-κB signaling pathway is crucial during development and inflammatory processes. We have previously shown that NF-κB activation induces dedifferentiation of astrocytes into neural progenitor cells (NPCs). Here, we provide evidence  that the NF-κB pathway plays also a fundamental role during the differentiation of NPCs into astrocytes. First, we show that the NF-κB pathway is essential to initiate astrocytic differentiation as its early inhibition induces NPC apoptosis and impedes their differentiation. Second, we demonstrate that persistent NF-κB activation affects NPC-derived astrocyte differentiation. Tumor necrosis factor (TNF)-treated NPCs show NF-κB activation, maintain their multipotential and proliferation properties, display persistent expression of immature markers and inhibit astrocyte markers. Third, we analyze the effect of  NF-κB activation on the main known astrocytic differentiation pathways, such as NOTCH and JAK-STAT. Our findings suggest that the NF-κB pathway plays a dual fundamental role during NPC differentiation into astrocytes: it promotes astrocyte specification, but its persistent activation impedes their differentiation.

## 1. Introduction

Several studies have shown the astrocytic ability to re-acquire neural progenitor cells (NPC) properties under specific conditions [1,2,3,4]. For example, mature parenchymal astrocytes are able to dedifferentiate towards earlier developmental stages upon injury and inflammation [2,5]. Thus, astrocytes are considered as potential and attractive candidates for cell-replacement therapies. Our previous studies indicate that NF-κB activation is a key element promoting astrocyte dedifferentiation [6]. Tumor necrosis factor (TNF)-induced NF-κB activation in astrocytes promoted loss of GFAP expression and markers related to glycogen metabolism coupled with re-expression of stemness markers, such as CD44, Musashi-1, and OCT4. In addition, we provided a role of OCT4 in this process of dedifferentiation [6]. As NF-κB activation promotes astrocyte dedifferentiation, would it be possible that the NF-κB pathway plays also a role during the astrocytic differentiation? In this context, the NF-κB pathway is considered as a master regulator in neurodevelopment. It is involved in cell proliferation, differentiation, cell survival, cell death, cell fate determination, and migration [7,8,9]. In particular, during early stages of brain development, the NF-κB pathway is selectively activated in neocortical neural progenitor cells to inhibit neuronal differentiation and maintain the progenitor cell pool [10]. The possibility that neuro-inflammation modulates the astrocytic differentiation during neural development is unknown, but has been suggested in a recent article [11] where it was shown that  acute in utero exposure to lipopolysaccharide induces inflammation in the pre- and postnatal brain and alters the glial cytoarchitecture in the developing amygdala. Although it is generally admitted that three major signaling pathways (Notch, STAT3, and BMP) are involved in astrocyte differentiation [12] and that cross-talk between these signaling and inflammatory pathways are currently observed [13,14,15], the role of the NF-κB pathway during astrocyte differentiation remains largely unknown. In our study, we have used a murine neurosphere model to investigate the potential role of the NF-κB pathway during the astrocytic differentiation. We identified two roles attributed, at least partially, to the NF-κB pathway during this process: its activation during astrocytic specification prevents the apoptosis of pre-astrocytic cells, while the persistent activation of the NF-κB pathway by TNF at later stages induces an inhibition of the astrocytic development via a modulation of Notch and STAT3 signaling. Thus, the NF-κB pathway appears to be a new key element in the differentiation process of astrocytes.

## 2. Materials and Methods

### 2.1. Ethic Statement

All animal procedures were conducted in strict accordance with the 2010/63/EU European Union Directive and with the local Committee for Care and Use of Laboratory Animals.

### 2.2. Cell Culture

Primary cultures of neurospheres (NSPs) were obtained from embryonic murine neural stem cells derived from the ventricular zone of embryonic day 14 (E14) C57BL/6J mouse embryos (Harlan, The Netherlands) as described previously [16]. NSPs were cultured in neurobasal medium (DMEM F12; Lonza, Basel, Switzerland) supplemented with 1% B27 without vitamin A (Life Technologies, Waltham, MA, USA), penicillin (100 U/mL; Lonza, Basel, Switzerland), streptomycin (100 g/mL; Lonza, Basel, Switzerland), and 20 ng/mL EGF (epidermal growth factor; Life Technologies, Waltham, MA, USA ).

NSPs were differentiated into astrocytes on poly-L-ornithine-pre-coated 6 wells plates with DMEM containing 10% FBS (fetal bovine serum; Gibco, Waltham, MA, USA), penicillin (100 U/mL; Lonza), and streptomycin (100 g/mL; Lonza). Cells were differentiated with and without TNF (50 ng/mL; R&D Systems, Minneapolis, Minn, USA) and then incubated for 6, 24, 48, or 72 h at 37 °C in 5% CO_2_.

### 2.3. NF-κB Inhibition

The NF-κB pathway was inhibited using JSH-23 (4-Methyl-N^1^-(3-phenylpropyl)benzene-1,2-diamine) (Calbiochem, Merck Millipore, Burlington, MA, USA). JSH-23 is a selective inhibitor of nuclear translocation of NF-κB p65 and its transcription activity.

### 2.4. MTT Cell Viability Assay

The MTT (3-[4,5-dimethylthiazol-2-yl]-2,5-diphenyltetrazolium bromide) assay is widely used to evaluate cytotoxicity and cell viability. Cells were seeded into 96-well culture plates and incubated with MTT (5 mg/mL) at 37 °C for 3 h. The medium was then removed and cells were lysed with dimethyl sulfoxide (DMSO; VWR, Radnor, PA, USA). MTT conversion was quantified by measuring absorbance at 540 nm using a microplate reader (TECAN, Mannedorf, Switzerland).

### 2.5. Tunel Assay

Cells were cultured on poly-L-ornithine coated coverslips in the presence of TNF (50 ng/mL) or JSH-23 during 6, 24, or 72 h. Fragmented DNA of apoptotic cells was revealed using Dead-End^TM^ Fluorometric TUNEL (Terminal deoxynucleotidyl transferase dUTP Nick End Labeling) according to the manufacturer’s protocol (Promega, Madison, WI, USA). Coverslips were mounted with DAPI-Fluoromount G and observed under a LSM 510 META inverted confocal microscope.

### 2.6. Western Blot

Total proteins were extracted with RIPA buffer (Pierce, Rockford, IL, USA) and 1% protease/phosphatase inhibitor cocktail (Pierce). Cell extracts were analyzed by immunoblot using rabbit anti-Caspase-3 (1:1000; cell signaling technology, Danvers, MA, USA #14220), rabbit anti-GFAP (1:1000; cell signaling technology, Danvers, MA, USA #12389), mouse anti-GAPDH (1:1000; Sigma, Saint Louis, MO, USA G8795), mouse anti-STAT3 (1:1000; BD, Allschwil, Switzerland #610189), and rabbit anti-P-STAT3 (1:1000; cell signaling technology, Danvers, MA, USA #9145). Analysis was done using the BioRad ChemiDoc™ XRS + System and the Image Lab software (Biorad, Hercules, CA, USA). Quantification of bands was performed using Image J software.

### 2.7. Microarrays Experiments and Quality Control

NSPs were treated with TNF (50 ng/mL) at 0 h for 6, 24, 48, and 72 h. After treatment, total RNA was extracted, using RNA NOW^TM^ reagent (Ozyme, Saint Cyr L’Ecole, France) according to the manufacturer’s instructions. Total RNA integrity and purity were assessed using the Agilent 2100 Bioanalyzer and RNA 6000 Nano LabChip kits (Agilent Technologies, Santa Clara, CA, USA). Microarrays analysis was performed with Affymetrix MoGene1.0 ST arrays as per manufacturer’s instructions. For each microarray experiment, the necessary quality controls were made. Ingenuity pathway analysis (IPA) was used to analyze the canonical and functional pathways implicated on some of our gene dataset as well as to explore pathways involved in the inflammatory response. Microarray expression data are available at the GEO database (GSE117736).

### 2.8. Real-Time PCR

Total RNA was extracted using the innuPREP RNA Mini Kit (Westburg, Leusden, The Netherlands) following manufacturer’s protocol. Complementary DNA (cDNA) was synthesized from RNA samples using the ImProm-II Reverse Transcription System (Promega, Madison, WI, USA). Primer sequences were designed using the Beacon Designer Software (Biorad, Hercules, CA, USA) and used at a final concentration of 500 nM (Table 1). Gene expression was analyzed using Bio-Rad iCycler (iQ5 Real-Time PCR Detection System, Biorad, Hercules, CA, USA) with SYBR Green supermix (Promega). β-actin was used as housekeeping gene. The 2−∆∆CT method was used to determine the relative changes of gene expression.

### 2.9. Immunocytochemistry

NSPs were cultured on poly-L-ornithine coated coverslips and then fixed with paraformaldehyde (4% in phosphate buffer saline (PBS)), followed by a permeabilization with 0.3% Triton X-100 in PBS. After washing in PBS, the blocking step was done in PBS containing 3% BSA at room temperature for 30 min. The cells were then incubated overnight at 4 °C with primary antibodies, purified mouse anti-human KI67 (1:100; BD Biosciences, Allschwil, Switzerland #550609), rabbit anti-CD133 IgG (1:100; Abcam, Cambridge, UK #ab19898), rabbit anti-CD44 IgG (1:100; Abcam, Cambridge, UK #ab243894), and rabbit anti- NF-κB p65 (1:100; cell signaling technology, Danvers, MA, USA #8242). After washing, cells were incubated with the corresponding anti-rabbit or anti-mouse secondary antibodies conjugated to cyanine 2 or cyanine 3 for 1 h at room temperature (1:1000; Jackson ImmunoResearch, Cambridge, UK). For the GFAP staining, a cyanine 3-conjugated mouse anti-GFAP IgG antibody (1:800; Sigma C9205) was used. Cells were then washed and mounted with DAPI-Fluoromount G (SouthernBiotech, Birmingham, AL, USA). The immunoreactivity was assessed using a LSM 510 META inverted confocal microscope (Carl Zeiss Micro Imaging, Oberkochen, Germany) and analyzed on ImageJ software.

### 2.10. Microarray, Limma Analysis

The *limma* R package enabled contrast comparison using linear models and moderated *t*-tests. Briefly, this method [17] computed several summary statistics via the *eBayes ()* function for each gene and each contrast. Compared to the classic *t*-statistic, the standard errors are moderated across all genes, thus shrunk to a common value using a Bayesian model. From moderated *t*-statistics, *p*-values are deduced as for ordinary *t*-statistics but degrees of freedom are increased. It reflects the greater reliability associated with the smoothed standard errors [18]. The four investigated contrasts were: Dif6hr = TNF_6h-FCS_6h, Dif24hr = TNF_24h-FCS_24h, Dif48hr = TNF_48h-FCS_48h and Dif72hr = TNF_72h-FCS_72h. Once the contrasts were obtained, to better control testing multiple comparisons, *q-values* were computed using the R package qvalue [19]. Gene annotations for the mouse expression platform *mogene-1.0* were downloaded from the affymetrix website, the version 36 was used (file MoGene-1_0-st-v1.na36.mm10.transcript.csv). All computation and plotting were performed using R {R Development Core Team, 2016 #1613} version 3.3.1 and RStudio (RStudio Team (2017). RStudio: Integrated Development for R. RStudio, Inc., Boston, MA, USA URL http://www.rstudio.com/, accessed on 5 April 2021).

### 2.11. Statistical Analysis

Data are represented as mean  ±  standard error of the mean (SEM). Statistical analyses are performed on a least 3 independent experiments. Multiple group comparisons were determined using Kruskal–Wallis test followed by Dunn’s post-test. A two-way ANOVA on gene expression, testing similarly the effect of TNF treatment and time, was performed on the delta-Ct values followed by a Tukey post hoc test to determine the source of the variation. Statistical analyses were carried out using Graphpad Prism software. Results were considered as significant when *p*-values were less than 0.05 (*), 0.01 (**), or 0.001 (***).

## 3. Results

### 3.1. NF-κB Is Essential for the Initiation of NPCs-Derived Astrocyte Differentiation

In order to investigate the implication of the NF-κB pathway during astrocyte differentiation, we performed loss of function experiments using JSH-23, an inhibitor of NF-κB nuclear translocation. Neurospheres (NSPs) were left untreated or treated at time 0 (when neurospheres were differentiated on poly-ornithine) with JSH-23 and cultivated in differentiation medium (DMEM containing 10% FBS) for 48 h. A prominent loss of viability occurred in cells exposed to JSH-23 at 1, 5, 10, and 20 μM compared to untreated NSPs (38.8%, 31.3%, 36%, and 4.3% viable cells, respectively) (Figure 1A). Next, NSPs were treated with JSH-23 (10 μM) at different time points along their differentiation (0, 24, 48, or 72 h) and maintained in differentiation medium until 96 h. JSH-23 induced high loss of cell viability when NSPs were treated at time 0, but cell viability was not affected in NSPs treated with JSH-23 at 24, 48, or 72 h (Figure 1B). To confirm these results, we used an alternative NF-κB inhibitor, BAY11-7082, with a different mode of action: it inhibits IKK activity thereby maintaining NF-κB in its inactive state [20]. NSPs were treated with BAY11-7082 (10 μM) at different time points along their differentiation (0, 24, 48, or 72 h) and maintained in differentiation medium until 96 h. As JSH-23, BAY11-7082 induced high loss of cell viability only when NSPs were treated at time 0, but not at later time points (Appendix A). Next, in order to investigate the mechanism responsible for NSPs loss of viability, we analyzed *caspase-3 (Casp3)* gene expression levels in JSH-23-treated NSPs at time 0 after 6 h in differentiation conditions. *Casp3* expression levels were upregulated after JSH-23 treatment at 10 and 20 μM when compared to untreated cells (Figure 1C). At the protein level, we confirmed increased levels of the cleaved form of CASP3 after 3 or 6 h of JSH-23 exposure (10 and 20 μM) when compared to control conditions (Figure 1C). In addition, TUNEL experiments of cells treated with JSH-23 after 3 and 6 h of differentiation illustrate the process of apoptosis induced by the inhibition of the NF-κB pathway by JSH-23 (Figure 1D). Taken together, these results show that the NF-κB pathway is essential for the initiation of NSPs-derived astrocyte differentiation, while it is not necessary at later stages.

### 3.2. TNF-Treated NPCs Show an Inflammatory Phenotype Associated to NF-κB Activation

To investigate if persistent activation of the NF-κB pathway along NPCs differentiation into astrocytes could lead to perturbation of the normal developmental processes, we treated NSPs with TNF (50 ng/mL) and cultivated cells in differentiation medium. In order to follow the differentiation process, we performed genome-wide gene expression analysis at 6, 24, 48, and 72 h and analyzed differentially expressed genes (DEGs) between untreated and TNF-treated NSPs (Figure 2). As expected, the NF-κB pathway was highly activated at all stages along NSPs differentiation. Genes associated to the NF-κB pathway, such as *Nfkbia*, were in the top 25 upregulated genes at all stages (Figure 2). We confirmed increased expression levels of *Nfkbia* (or *IκBα*) in TNF-treated NSPs compared to untreated cells by qPCR (Appendix A). Inflammatory-associated genes classically upregulated in reactive astrocytes, such as *lipocalin 2* (*Lcn2*), *chemokine (C-C motif) ligand 2* (*Ccl2*), *Ccl7*, and *toll-like receptor 2* (*Tlr2*), were also among the top 25 upregulated genes upon TNF exposure when compared to untreated NSPs (Figure 2). These observations demonstrate that TNF-treated NSPs possess an inflammatory signature through NF-κB κB activation. To check if a TNF treatment induced apoptosis along the differentiation process, we performed a TUNEL assay. To this end, we differentiated NSPs during 6, 24, and 48 h with and without TNF. We observed more TUNEL labeled cells in the presence of TNF (Appendix A) and cell death was increased at 6, 24, and 48 h (30.6%, 21.9%, and 28.0% respectively) (Appendix A). However, it is important to note that, together with apoptosis, we observed an increase of the proliferation rate following a TNF treatment. Thus, at the end of the differentiation process, we measured similar number of cells in control or TNF conditions. This was further reflected by similar amounts of RNA levels that we measured between the two conditions (data not shown).

### 3.3. NF-κB Activation Following TNF Exposure Increases Immaturity Markers and Modulates Astrocytic Differentiation

Next, following the observation that TNF-treated NPCs show an inflammatory phenotype through NF-κB activation at all stages along development, we aimed to investigate the potential concomitant effect of this persistent inflammatory status on NSPs-derived astrocyte differentiation. For this, we analyzed the effect of TNF on immaturity markers, including the neural progenitor marker CD133 and the pre-astrocyte marker CD44. Interestingly, CD133 expression levels were upregulated following TNF treatment, although, because of the high variability between the experiments, differences were not significant (Figure 3A,C). At the protein level, immunocytochemistry staining revealed that CD133 expression persisted at 24 h under TNF condition compared to the control conditions, where CD133 expression levels was largely decreased (Figure 3B). However, after 72 h of differentiation, CD133 expression levels were decreased in both conditions (Figure 3B). Further, TNF exposure increased CD44 expression levels at 6 and 24 h, while its levels were decreased at 48 and 72 h when compared to untreated cells (Figure 3D). At the protein level, CD44 expression was upregulated at 72 h in TNF-treated cells when compared to untreated cells, thus suggesting a post transcriptional regulation of CD44 under inflammatory conditions (Figure 3D). We then focused on the expression levels of classical astrocyte markers. The expression of glial fibrillary acidic protein (GFAP) peaked at 72 h both at the transcriptional and protein level (Figure 4A,C). Following a TNF treatment, GFAP expression levels were highly decreased at 72 h when compared to control conditions (Figure 4B–D). In line with these results, aldehyde dehydrogenase family 1, subfamily A1 (Aldh1a1), a marker of astrocytic differentiation during brain development [21], was in the top 25 down-regulated genes upon TNF exposure at 24, 48, and 72 h (Figure 2). Interestingly, through the microarray datasets, we observed that astrocytic genes described in the literature and expressed during astrocyte differentiation, were down-regulated upon TNF exposure (Appendix A). Taken together, these results show that persistent NF-κB activation during NPCs differentiation induces the maintenance of NPC markers as well as impedes the expression of astrocytic markers.

### 3.4. TNF-Treated NPCs Display Increased Proliferation and Maintain Multipotential Properties along Astrocyte Differentiation

Next, we explored the effect of NF-κB activation on cell proliferation by analyzing *mKi67* expression levels along the differentiation stages. Its expression decreased significantly during differentiation, already at 24 h (Figure 5A). Following a TNF exposure, *mKi67* gene expression was significantly enhanced at 24 h (Figure 5B). Gene expression results were corroborated at the protein level, the latest showing upregulation of Ki67 at 24 h in TNF-treated cells when compared to untreated cells (Figure 5C). However, Ki67 expression was not detected in both conditions after 72 h (Figure 5C). In order to investigate if NSPs-derived astrocytes treated with TNF exhibit NPC properties, we performed a NSP reformation assay. After differentiating NSPs in the presence or absence of TNF during 6, 24, or 48 h, we further cultivated the cells in NSP reformation medium (containing EGF and FGF at 20 ng/mL) and counted, after 10 days, the number of formed NSPs. After 6 and 24 h of differentiation in control conditions, cells were still able to form NSPs (36.2 and 32.8, respectively). However, this capacity was highly reduced after 48 h of differentiation (6.3) (Figure 5D). When differentiating cells were exposed to TNF during 6, 24, or 48 h, the number of reformed NSPs was higher when compared to control conditions (44.3, 63, and 18.7, respectively). Taken together, these results highlight that inhibition of differentiation induced by TNF stimulation could be correlated with an increase of glial progenitor proliferation and that NF-κB activation induces the maintenance of multipotential properties typical of undifferentiated.

### 3.5. NOTCH and JAK-STAT Signaling Pathways Are Modulated in TNF-Treated NPCs-Derived Astrocytes

After having shown that TNF is able to modulate the differentiation of NSP-derived astrocytes, we aimed to identify the signaling pathways implicated in this process. To this end, we studied the effect of NF-κB activation on NOTCH, JAK/STAT, and BMP signaling, three fundamental pathways implicated in astrocytes development.

First, we focused on the Notch pathway and analyzed the expression of its downstream effectors HES1, MASH1, and NEUROD1 known to inhibit glial differentiation. *Hes1* expression was significantly down-regulated in presence of TNF at 48 and 72 h (Figure 6A). In TNF-treated cells, *Mash1* expression was highly expressed at 6 and 24 h, while it was not affected at later time points (Figure 6B). Concerning the pro-neuronal factor NEUROD1, its corresponding gene expression levels were significantly upregulated at 24 h in TNF-treated cells when compared to untreated cells (Figure 6C). Next, in order to analyze the effect of TNF on the STAT3 pathway, known to promote astrocyte differentiation, we studied the expression level of SOCS3, the direct effector of the pathway. *Socs3* expression was significantly and constantly increased over time during NSPs-derived astrocyte differentiation (Figure 6D). Following a TNF treatment, *Socs3* expression was significantly decreased at 72 h of differentiation (Figure 6E). In addition, we have monitored phosphorylated STAT3 protein during differentiation (Appendix A). We showed that p-STAT3, which reflects the activation of the pathway, increases after 24 h of differentiation and that TNF treatment had no effect on STAT3 phosphorylation levels (Appendix A). Thus, the decrease of downstream target of JAK-STAT pathway as SOCS3 and GFAP, reflects that TNF inhibits STAT3 function, but as showed by Western blot, this is not due to STAT3 phosphorylation, but probably via an interaction of subunits of the NF-κB transcription complex and p-STAT3 into the nucleus.

Finally, we investigated the effect of NF-κB activation on the BMP pathway, which is also known to be implicated in the differentiation of astrocytes. We analyzed the expression levels of *Id1* and *Id3*, corresponding to the direct target genes of the BMP pathway. Following a TNF treatment, we observed a significant decrease of *Id1* at 48 h and a slight non-significant decrease of *Id3* at 48 h and 72 h (Appendix A). These results suggest that if TNF treatment induces a down regulation of Notch and STAT3 pathways, NF-κB activation the modulation of BMP pathway.

## 4. Discussion

Astrocytes are crucial players in the developing and adult CNS in both health and disease. Consequently, it is critical to elucidate the mechanisms underlying acquisition of astrocyte identity and plasticity, as well as understanding how these processes are modulated under pathological conditions [22]. Evidence accumulated in the past years demonstrates that astrocytes exhibit a functional and morphological degree of plasticity, which are notably observed in lesions. Several reports investigated the ability of astrocytes to dedifferentiate into neural progenitor cells (NPCs) in response to tissue damage [2,3]. Therefore, our recent results showing that tumor necrosis factor (TNF), which triggers NF-κB activation, was able to convert primary astrocytes into immature progenitors are in line with these reports [6]. In the central nervous system (CNS), TNF is the main inducer of the NF-κB pathway and it is among the key proinflammatory cytokines mainly produced by microglia as well as by other cell types, such as brain-resident astrocytes or neurons, in response to insults [23]. These findings highlight the possible role of the NF-κB signaling pathway in gliogenesis and, in particular, during astrocyte differentiation. The NF-κB signaling pathway is known to play an important role in inflammation as well as in developmental processes [24,25,26]. Its essential role during development has been specifically demonstrated in a neurogenic context, such as neurogenesis, neuronal transmission, and neuroprotection [27,28,29,30,31]. Recently, the complex involvement of NF-κB proteins in different aspects of postnatal neurogenesis has been reported [32,33]. In addition, in a human cellular model of neuronal differentiation, a novel essential function of NF-κB-c-REL in fate choice of NSCs between neurogenesis and oligodendrogenesis has been described. c-REL inhibition led to a significant increase of apoptosis, while the surviving cells switched their fate and differentiated into oligodendrocytes instead [34]. However, our current understanding of the role of NF-κB signaling on glial cells differentiation remains limited. For this reason, in the present study, we investigated the role of the NF-κB signaling pathway on astrocytic differentiation through loss and gain of function experiments. We took advantage of the well-established neurospheres (NSPs) model, which, under specific culture conditions, retain multipotential properties, thus being able to give rise to neurons, astrocytes, and oligodendrocytes. For our experiments, we differentiated neurospheres into astrocytes by cultivating them with fetal bovine serum, as previously described [35,36]. We inhibited the NF-κB pathway by JSH-23, which has been described as the nuclear translocation inhibitor of p65/p50 dimer of NF-κB [37]. Inhibition of this pathway in NPCs by JSH-23 prevented their differentiation into astrocytes and induced apoptosis. We confirmed these results by treating the cells with BAY11-7082, an alternative NF-κB inhibitor [20]. To our knowledge, we were the first to provide evidence of the crucial role of the NF-κB pathway during the initiation of astrocyte differentiation. As JSH-23 inhibits nuclear translocation of p65/p50 dimer of NF-κB, we could hypothesize that RELA/p65 could be the main effector of the inhibition of astrocytic apoptosis during the first 24 h of astrocyte differentiation. It is interesting to note that during glutamatergic differentiation of hNSCs originating from neural crest-derived stem cell, the NF-κB subunit distribution shows that levels of nuclear p65 protein, revealed a further significant decrease during early neuronal differentiation while a significant increase of nuclear c-REL protein was detected on day 2 of neuronal differentiation [34]. These observations are in accordance with the decrease of p65 in the nucleus observed during the first 24 h of astrocyte differentiation (Appendix A). Several studies have suggested the dual role of NF-κB as an attenuator or promoter of apoptosis [38]. Inhibition of NF-κB in immature murine B-cells has been previously shown to result in apoptosis [39]. In neural cells, the RelA subunit has been demonstrated to contribute to neuronal cell death, while the overexpression of c-Rel factor protects from cell death [40,41]. Further, Bellavia and collaborators have demonstrated that the expression of a constitutive form of Notch3 leads to specific alterations in thymocytes development. The authors described a constitutive activation of NF-κB in malignant T cells from Notch3 transgenic mice. Additionally, these mice showed activation of anti-apoptotic pathways NF-κB dependent in T-lymphoma cells [42]. In this context, it is interesting to note that in the first stage of astrocyte differentiation corresponding to the stage of specification, activation of the NOTCH pathway inhibits the expression of pro-neural genes and induces acquisition of astrocytic fate [16]. We hypothesize that at early stages of NPCs differentiation in our model, the NOTCH pathway induces a constitutive activation of NF-κB, which prevents apoptosis of pre-astrocytes.

Next, to investigate if persistent activation of the NF-κB pathway along NPCs differentiation into astrocytes could lead to perturbation of the normal developmental processes, we treated NSPs with TNF during astrocyte differentiation and analyzed their transcriptional signature at different time points. Our transcriptomic results showed the activation of the NF-κB pathway, which was highly regulated at all times following TNF exposure. We observed that TNF induced NSPs proliferation, showing an increase of Ki67 expression compared to untreated cells. Interestingly, Kaltschmidt and collaborators found similar results with bromodeoxyuridine (BrdU) staining [28].

Next, we aimed to investigate the effect of a persistent inflammatory status on NSPs-derived astrocyte differentiation. We reported that TNF favors the maintenance of astrocytic precursors, such as CD44, a pre-astrocytic marker [43], and CD133, a marker of neural progenitors [44]. Additionally, we showed that NSPs-derived astrocytes treated with TNF were able to reform neurospheres. In parallel, astrocytic classical markers, such as *Gfap*, *Aldh1a1*, and different genes expressed during astrocyte differentiation were down-regulated upon TNF treatment, confirming that the astrocytic differentiation was affected under inflammatory conditions. We hypothesized that NF-κB activation directs astrocyte differentiation towards a reactive glial phenotype. However, the effect of TNF was not fully reflecting a reactive astrogliosis phenotype, generally defined by an upregulation of GFAP expression, while in our hands, its expression was decreased. This contradiction could be explained by the fact that mature astrocytes that become reactive do not express GFAP under resting conditions [45]. Additionally, the role of GFAP could be different during reactive astrogliosis of mature astrocytes and in astrocytes which differentiate.

To understand the mechanisms that induced the modulation of astrocyte differentiation by TNF, we focused on the main signaling pathways implicated in this process. We showed that NF-κB activation has an effect on the NOTCH pathway, where *Hes1* expression was significantly downregulated, while *Hes5* (Appendix A) expression was upregulated. These results are in line with the fact that HES1 is known to regulate the astrocytic cell fate while HES5 maintains glial cells at a precursor stage [46]. These observations suggest that NF-κB activation inhibits the astrocytic cell fate through a decrease of *Hes1* and favors the state of glial precursors through the expression of *Hes5*. In addition, the pro-neuronal genes *Mash1* and *NeuroD1* were upregulated, suggesting that TNF could favor neuronal development. These findings can be related to the study of Heinrich and collaborators who described that astrocytes can be reprogrammed in vitro to generate neurons following injury [47]. It was also shown that both NOTCH and NF-κB signaling pathways may contribute by reciprocal crosstalk to regulate neurodevelopment in physiological and pathological conditions. Different cellular contexts showing the interaction of NOTCH and NF-κB signaling pathways have been described [48,49]. A recent study indicates that the lack of NF-κB p50 subunit, via dysregulation of other key neurodevelopmental pathways, such as NOTCH1, is associated with neurodevelopmental disorders [33]. We also showed that NF-κB activation down-regulated the STAT3 pathway after 48 h of differentiation when the signaling is normally activated. It is well-known that the STAT3 pathway induces astrocytic differentiation by activating astrocyte-specific genes such as *Gfap* [50]. GFAP expression was reduced at 48, 72 h, and 1 week after TNF treatment compared to untreated cells. This decrease could be potentially linked to the inhibition of STAT3 pathway after 48 h. The roles of NF-κB and STAT3 in colon, gastric, and liver cancers have been extensively investigated and the activation and interaction between STAT3 and NF-κB plays a fundamental role in the control of the communication between cancer cells and inflammatory cells [14,51].

For the BMP pathway, we found and confirmed that *Id1* and *Id3* expression levels were significantly expressed during normal NSPs-derived astrocyte differentiation [52]. Following a TNF treatment, we observed a significant decrease of *Id1* at 48 h and a slight non-significant decrease of *Id3* at 48 h and 72 h (Appendix A). These reductions could explain, at least partly, the downregulation of GFAP expression upon TNF treatment confirming that the BMP pathway induces NSPs differentiation into GFAP positive cells [53]. These results are also in line with the described downregulation of *Id1* and *Id3* expressions in astrocytes cultivated in dedifferentiation medium containing TNF [4]. It is plausible that NF-κB activation represses the 3 main pathways driving astrocyte differentiation and that this effect is also due to epigenetic modifications. The effect of NF-κB activation on specific histone marks associated with open or closed chromatin at specific loci in dedifferentiating astrocytes has been recently described [4].

In summary, the combination of our results on the role of the NF-κB pathway in the differentiation of astrocytes together with the results available in the literature enabled us to propose a model of astrocyte differentiation (Figure 7). At the stage of astrocyte specification characterized by the inhibition of pro-neural genes by the NOTCH pathway, the NF-κB signaling prevents the apoptosis of pre-astrocytic cells. Once the NF-κB pathway is down-regulated, astrocytes are able to differentiate in response to JAK/STAT and BMP signals and express their typical marker GFAP. Thus, our data suggest that the NF-κB signaling plays a fundamental role in glial specification and its persistent activation affects key developmental pathways, such as NOTCH and JAK/STAT signaling. This dialog between inflammatory and developmental pathways could have important implications on therapeutic strategies in the injured CNS to induce neuronal regeneration, but might also have implication in the context of brain tumors where inflammatory signals favor immaturity and proliferation of glioma cells.

## Figures and Tables

**Figure 1 cells-10-00840-f001:**
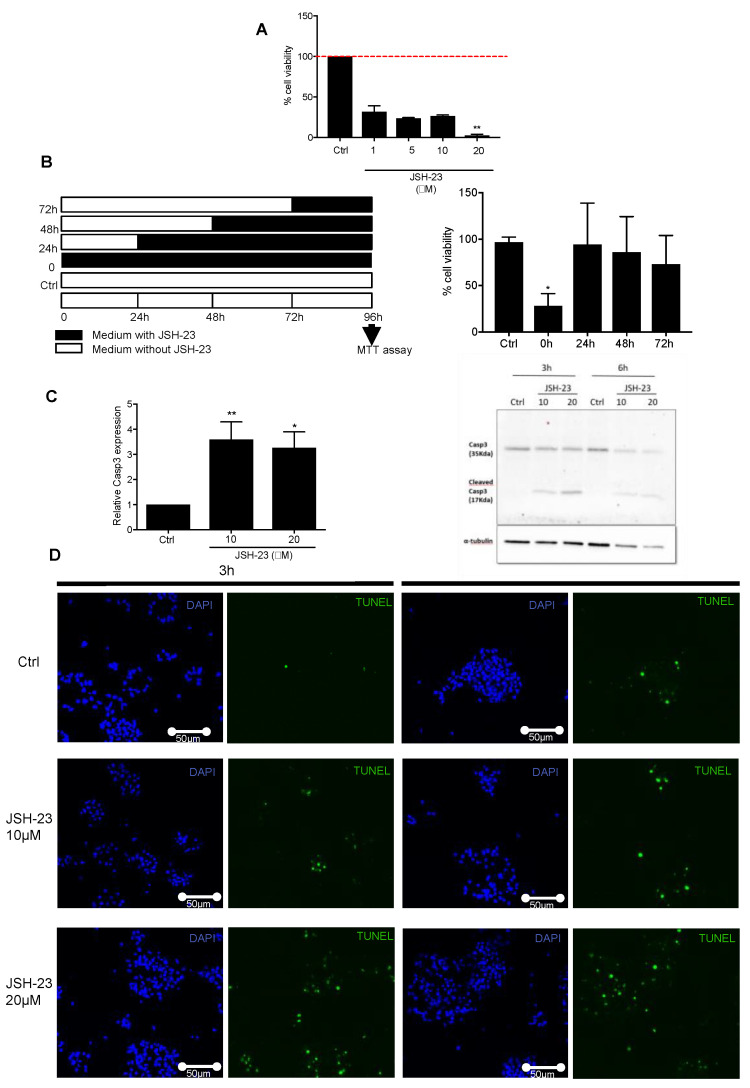
Inhibition of the NF-kB pathway by JSH-23 obstructs the astrocytic differentiation via Caspase 3 cleavage. (**A**) JSH-23 (1, 5, 10, 20 mM) induces the loss of cell viability by MTT assay after a 48 h treatment (*n* = 3, mean ± SEM). (**B**) JSH-23 decreases the cell viability during the first step of astrocyte differentiation. JSH-23 (10 mM) was added at different time points during astrocytic differentiation (0, 24, 48, 72 h) and the MTT assay was performed after 96 h (*n* = 3, mean ± SEM). * *p* < 0.05 (**C**) Analysis of the expression levels of Caspase3 after treatment with JSH-23 at different concentrations by RT-PCR (6 h) (C, *n* = 5, mean ± SEM) and by Western blot (3 and 6 h) (D, *n* = 3). α-tubulin was used as a loading control. * *p* < 0.05 and ** *p* < 0.01 (**D**) TUNEL assay was performed to explore cell death induced by JSH-23 (10, 20 mM) at 3 and 6 h. Nuclei were stained with DAPI (blue) and TUNEL positive cells display green nuclei. Scale bar: 50 µm.

**Figure 2 cells-10-00840-f002:**
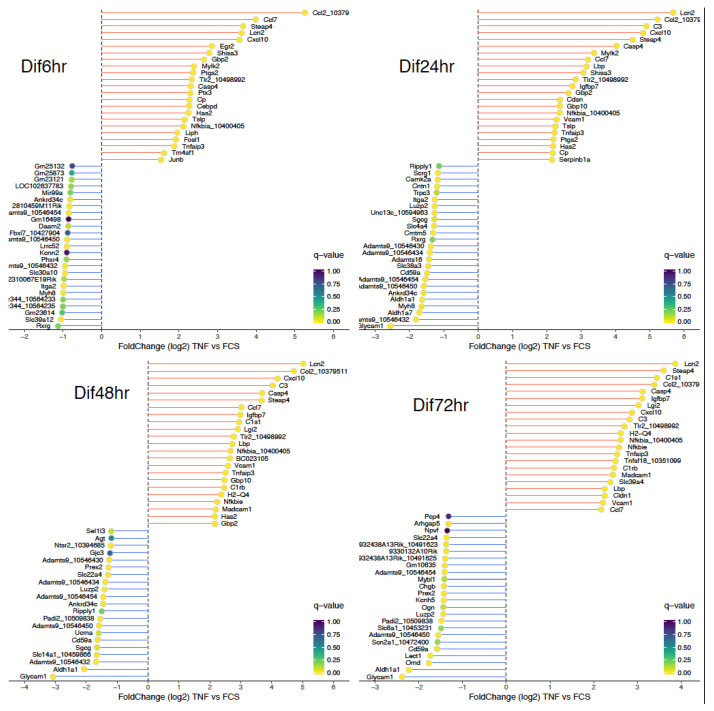
Snake plot. Top 25 and bottom 25 genes were extracted according to the log_2_ fold change value and plotted on the *y*-axis. Each gene dot is filled with its *q*-value and using the viridis color gradient (viridis R package, GarnierSversion0.4.0. https://CRAN.Rproject.org/package=viridis, accessed on 5 April 2021). Gene symbols are reported next to each dot.

**Figure 3 cells-10-00840-f003:**
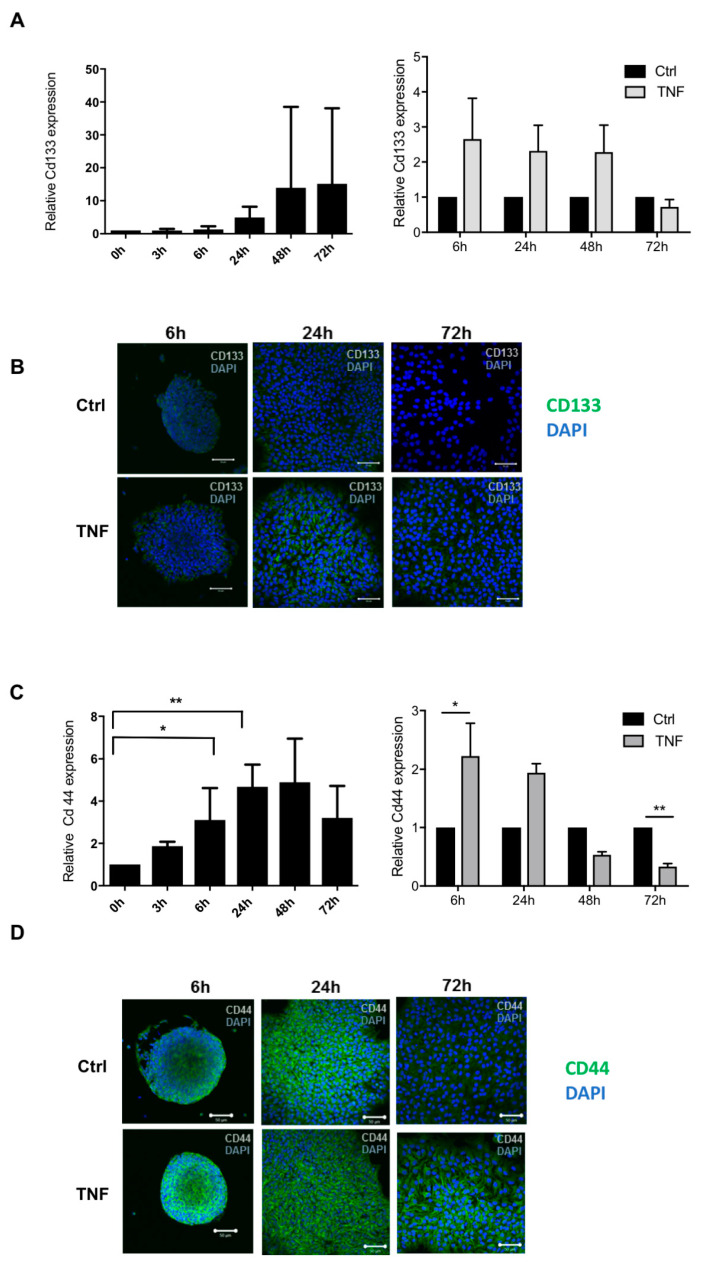
TNF tends to increase immaturity markers. Kinetic of Cd133 (**A**) and Cd44 (**C**) gene expression obtained by RT-PCR. Each time point is normalized to its 0 h of differentiation. (**A**) Effect of TNF treatment on Cd133 (**A**) and Cd44 (**C**) gene expression obtained by RT-PCR. Each time point is normalized to its FBS control (Ctrl = 1) * *p* < 0.05 and ** *p* < 0.01. Results are given as mean ± SEM (*n* = 4). (**B**) Immunocytochemistry showing CD133 (**B**) and CD44 (**D**) protein expression (green) in TNF treated cells at 6, 24, and 72 h. Nuclei were counterstained with DAPI (blue). Scale bar: 50 µm.

**Figure 4 cells-10-00840-f004:**
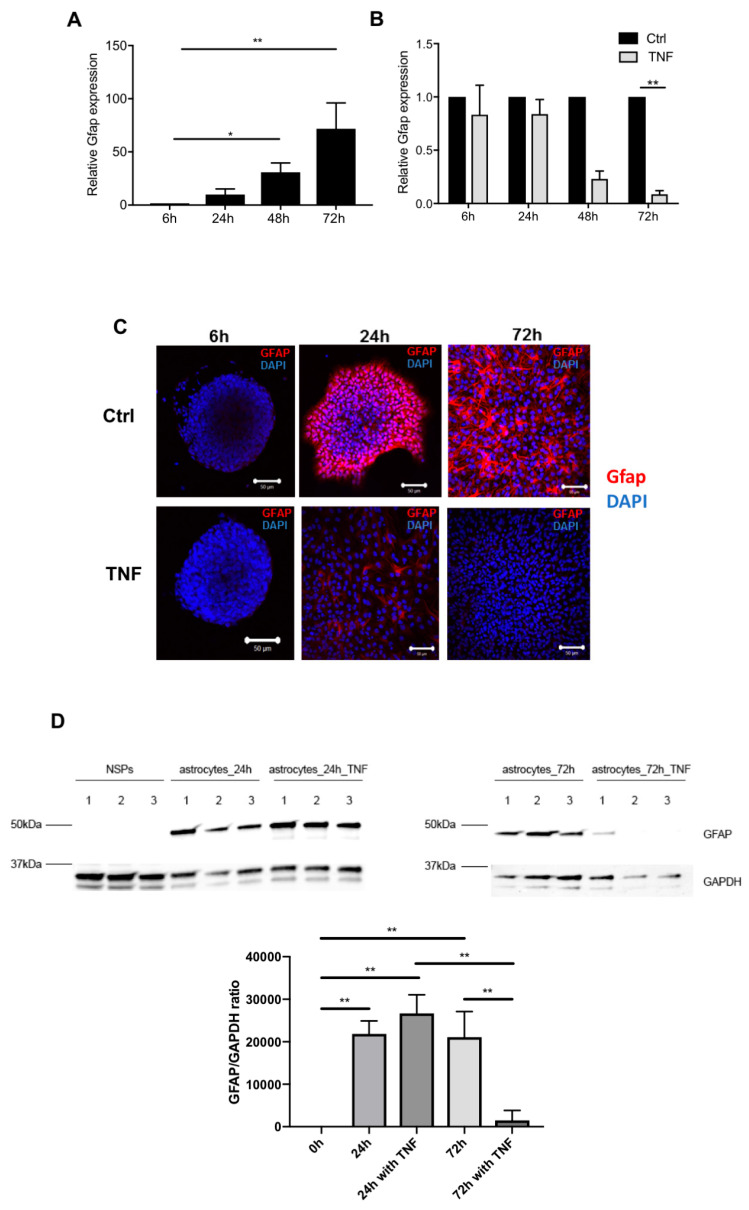
TNF modulates astrocytic differentiation through the classical Gfap marker. (**A**) Kinetic of Gfap mRNA gene expression obtained by RT-PCR. Each time point is normalized to its 6 h of differentiation. Results are given as mean ± SEM (*n* = 3) * *p* < 0.05 and ** *p* < 0.01. (**B**) Effect of TNF treatment on Gfap gene expression obtained by RT-PCR. Each time point is normalized to its FBS control (Ctrl = 1) ** *p* < 0.01. Results are given as mean ± SEM (*n* = 4). (**C**) Immunocytochemistry showing GFAP protein expression (red) in TNF treated cells at 6, 24, and 72 h. Nuclei were counterstained by DAPI (blue). Scale bar: 50 µm. (**D**) Immunoblots from TNF treated and untreated at 0, 24, and 72 h of differentiation. GFAP protein expression is affected by TNF, ** *p* < 0.01.

**Figure 5 cells-10-00840-f005:**
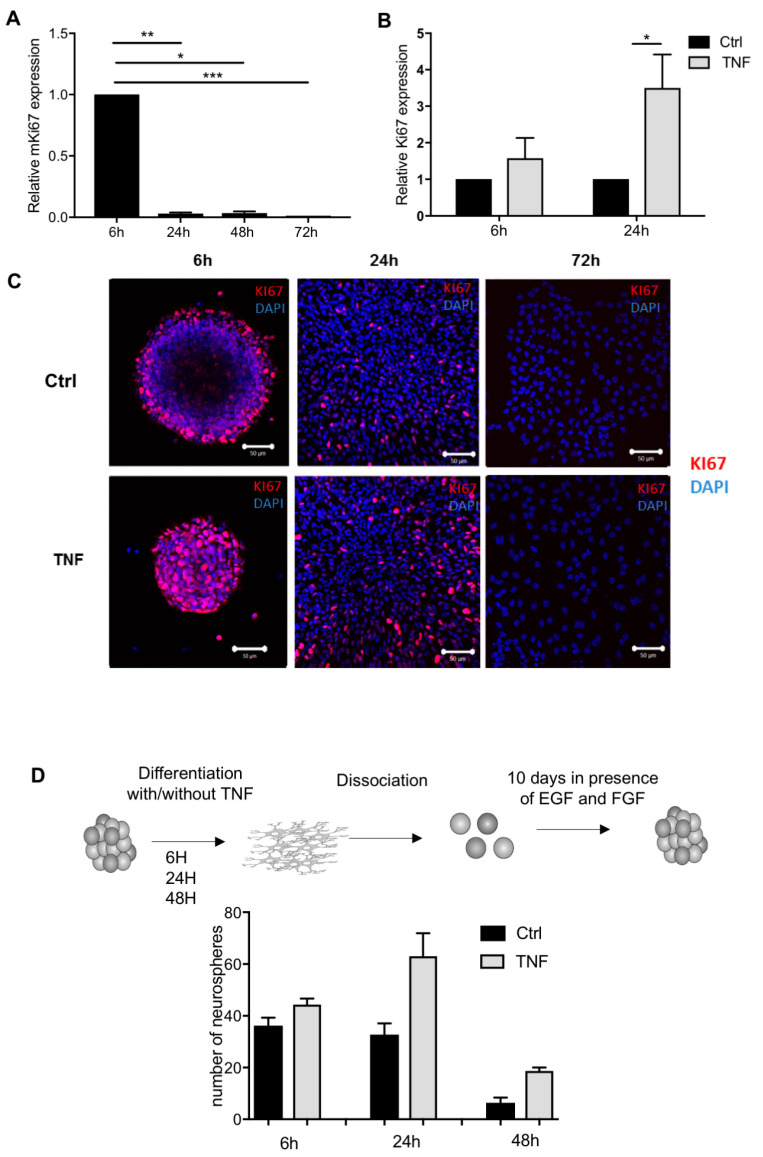
TNF increases neural progenitor cell proliferation. (**A**) Kinetic of mKi67 mRNA gene expression obtained by RT-PCR. Each time point is normalized to its 6 h of differentiation (Ctrl = 1). Results are given as mean ± SEM (*n* = 3) * *p* < 0.05, ** *p* < 0.01 and *** *p* < 0.001. (**B**) Effect of TNF treatment on mKi67 gene expression obtained by RT-PCR. Each time point is normalized to its FBS control (Ctrl = 1) * *p* < 0.05. Results are given as mean ± SEM (*n* = 4). (**C**) Immunocytochemistry showing KI67 protein expression (red) in TNF treated cells at 6, 24, and 72 h. Nuclei were counterstained by DAPI (blue). Scale bar: 50 µm. (**D**) TNFα induces the NSP reformation. After 6, 24, and 48 h of NSP-derived astrocyte differentiation treated with TNFα (50 ng/mL), an NSP reformation assay was performed in presence of EGF and FGF (20 ng/mL both, 10,000 cells/well). Ten days after, the number of reformed NSP at 6, 24, and 48 h was counted (*n* = 3, mean ± SEM).

**Figure 6 cells-10-00840-f006:**
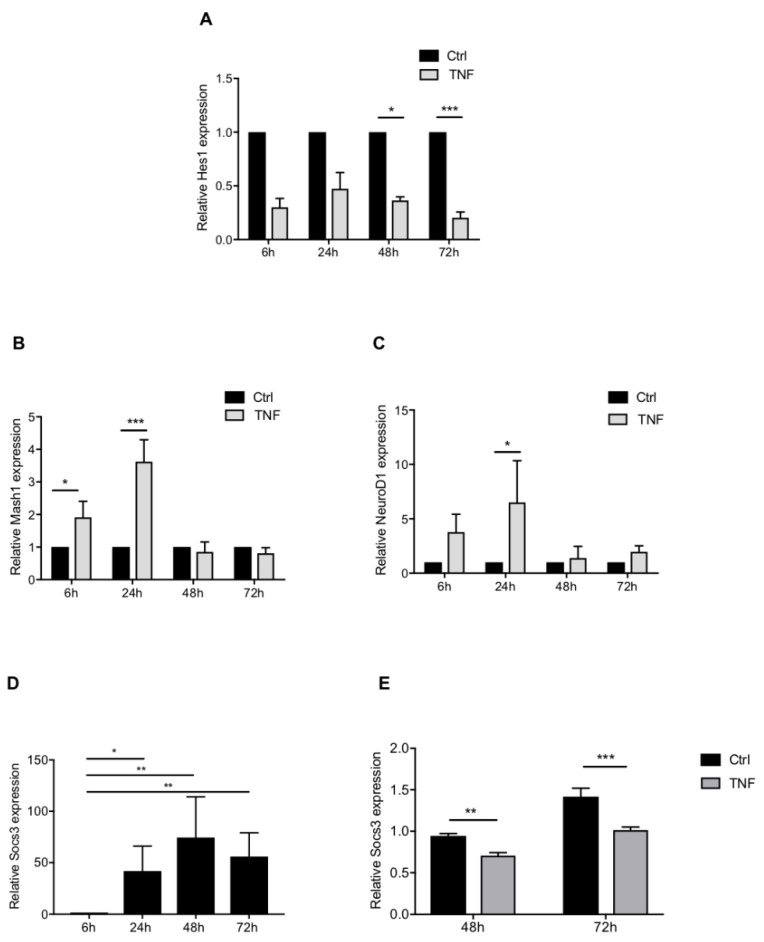
Effect of NF-kB activation on the main astrocytic signaling pathways during NSPs differentiation into astrocytes. Effect of TNF treatment on Hes1 (**A**), Mash1 (**B**), and NeuroD1 (**C**) genes expression obtained by RT-PCR. Each time point is normalized to its FBS control (Ctrl = 1). Results are given as mean ± SEM (*n* = 4). Kinetic of Socs3 mRNA gene expression obtained by RT-PCR. Each time point is normalized to 6 h of differentiation. Results are given as mean ± SEM (*n* = 3) (**D**). Effect of TNF treatment on Socs3 gene expression obtained by RT-PCR. Results are given as mean ± SEM (*n* = 6) (**E**). * *p* < 0.05, ** *p* < 0.01 and *** *p* < 0.001.

**Figure 7 cells-10-00840-f007:**
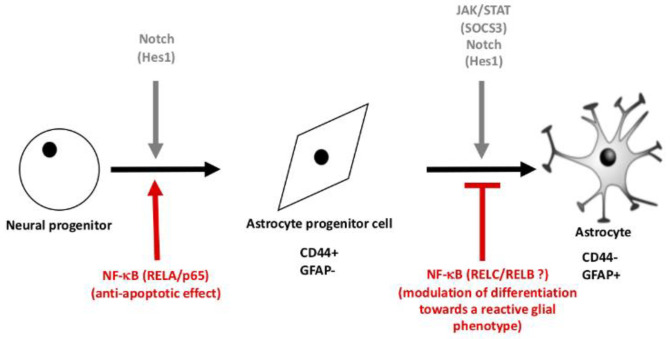
Proposed model for the role of NF-kB in the generation of astrocyte during neural stem cell differentiation.

**Table 1 cells-10-00840-t001:** List of primer sequences.

Gene	Forward	Reverse
*Casp3*	*GGCATTGAGACAGACAGT*	*GTAGAGTAAGCATACAGGAAGT*
*Cd133*	*ACCAGCGGCAGAAGCAGAATG*	*TGAGCAGACAAATCACCAGGAGAG*
*Cd44*	*TGGCACTGGCTCTGATTC*	*GTCTCTGATGGTTCCTTGTTC*
*Gfap*	*GGTTGAATCGCTGGAGGAG*	*CTGTGAGGTCTGGCTTGG*
*Hes1*	*GCCAATTTGCCTTTCTCATCC*	*GGTGACACTGCGTTAGGAC*
*Hes5*	*CGGTGGTGGAGAAGATGC*	*CTTGGAGTTGGGCTGGTG*
*Id1*	*TGCTGCCCTGATTATGA*	*GAAAGTCACCTTCCTGTAAA*
*Id3*	*ATGAACGGCTGCTACTC*	*CTCCACCTTGCTCACTT*
*I* *κBα*	*GCCAGTGTAGCAGTCTTGAC*	*GCCAGGTAGCCGTGAGTAG*
*Mash1*	*AAGATGAGCAAGGTGGAG*	*AGTCGTTGGAGTAGTTGG*
*MKi67*	*TTCCTTCAGCAAGCCTGAG*	*GTATTAGGAGGCAAGTT*
*NeuroD1*	*GAACTACATCTGGGCTCTG*	*GAAAGTCCGAGGGTTGAG*
*Socs3*	*ACCCTCCACAT-CTTTGTC*	*TCATACTGATCCAGGAACTCC*
*β-Actin*	*AGGGAAATCGTGCGTGACATCAAAGAG*	*GGAGGAAGAGGATGCGGAGTGG*

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
