# Peer review of "NF-κB and TNF Affect the Astrocytic Differentiation from Neural Stem Cells"

_cells, 2021, doi:10.3390/cells10040840_

Round 1
Reviewer 1 Report
The present manuscript adresses the role of NFκB-mediated signalling for the differentiation of neural progenitor cells (NPCs) to astrocytes. In this context, authors set a main focus on the TNF-α induced mechanisms and on signalling pathways, which are relevant for the differentiation process including NOTCH, STAT3 and BMP. According to the author’s hypothesis NFκB-mediated signalling maintains the initial proliferation status of NPCs, but impedes their further differentiation to astrocytes. Therefore, NFκB activation is important for the initiation phase of the differentiation process.
In general, evaluating the mechanisms of astrocyte differentiation and dedifferentiation is important regarding their potential role for cell-replacement therapies. To date, the role of NFκB in the context of astrocyte differentiation is largely unknown.
However, the provided information in this manuscript is more generally focussed on TNF-α mediated effects than on NFκB-dependent mechanisms. Therefore, the data do not provide enough evidence that NFκB is a main modulator of the mechansims analyzed in this work. I would recommend to include TNF-α also in the title of the manuscript. Furthermore, the correlations between the provided results are not clear enough and need to be pointed out more. In particular, the analyzed signalling pathways (NOTCH, STAT3, BMP) are involved a kind of randomly and their relations to NFκB are not provided experimentally. According to the provided figures the only common issue is the dependance on TNF-α. Working out more the relevance of NFκB-mediated signalling would improve the manuscript with respect to the title.
The following points can be adressed in further studies as they would improve the value of the manuscript:
1) Section 3.1: Figure 1 does not show data concerning changes in differentiation, it rather analyzes apoptosis levels after longer periods of inhibitor treatment, which may be unspecifically toxic to the cells in this case. Alternatively, siRNA experiments can be performed. However, if NFκB is essential for the initiation phase, one would assume that taking away the inhibitor from the cells after 24h of differentiation by a medium change would also block the differentiation but probably enhance cell viability? Furthermore, determining the level of apoptosis by Casp3 gene expression should be replaced by immunoblot analysis of the cleavage product of caspase 3 or of PARP cleavage product. Moreover, in figure 1D also TUNEL staining of later time points of the differentiation process should be shown, as according to the author’s hypothesis after 24, 48 or 72h of cultivation the addition of the inhibitor (for 3 and 6h) does not have an impact on differentiation, therefore one would not expect anymore TUNEL positive cells. This would be important to demonstrate that the impact of NFκB inhibition on survival occurs only at the initiation phase of differentiation, which is a main focus of this work. Finally, TUNEL positive cells should be quantified.
2) Section 3.2: As treatment with 50 ng/ml of TNF, which appears to be a very high dose, induces appoptosis in most cell types during the analyzed time points, cell viability should also be checked here as a control.
3) Section 3.3: In figure 3A and C do not provide enough evidence concerning the expression level of CD133 or CD44 transcripts over the time of differentiation following TNF-α treatment. For this, one should set the untreated control at time point 0h as 100% and then relate the data to this. This is important because values from untreated samples could also vary, which would have an impact on TNF-α values when the different time points are compared. The same applies for figure 4B. Furthermore, immunocytochemistry is not an adequate tool to quantify protein amounts, but more recommended for localization studies (fig. 3B, D). Figure 4D is more convincing.
4) Section 3.4: When comparing figure 5A and B one could see that after 24h the TNF-α induced Ki67 transcript values are apperently much lower than after 6h due to the abolished Ki67 expression (Fig. 5A) in untreated samples. Therefore, figure 5B provides a wrong impression and a one-way ANOVA test would probably erase significancies. Data from figure 5B should be integrated in figure 5A. Moreover, authors postulate that there is a correlation between induction of differentiation and inhibition of proliferation. Regarding the effects of TNF-α in this context, can they be reversed by adding the inhibitors? This would enhance the value of the manuscript as they would reveal NFκB-specific influences on proliferation, which are not clear in this section, yet.
Author Response
In general, evaluating the mechanisms of astrocyte differentiation and dedifferentiation is important regarding their potential role for cell-replacement therapies. To date, the role of NFκB in the context of astrocyte differentiation is largely unknown.
We thank the reviewer for highlighting the importance of our studies and the constructive comments on the manuscript.
However, the provided information in this manuscript is more generally focussed on TNF-α mediated effects than on NFκB-dependent mechanisms. Therefore, the data do not provide enough evidence that NFκB is a main modulator of the mechansims analyzed in this work. I would recommend to include TNF-α also in the title of the manuscript. Furthermore, the correlations between the provided results are not clear enough and need to be pointed out more. In particular, the analyzed signalling pathways (NOTCH, STAT3, BMP) are involved a kind of randomly and their relations to NFκB are not provided experimentally. According to the provided figures the only common issue is the dependance on TNF-α. Working out more the relevance of NFκB-mediated signalling would improve the manuscript with respect to the title.
We thank the reviewer for these suggestions. In the revised version of the manuscript we have changed the title of the article to reflect more its contents. In fact, we have realized loss and gain of NFκB function during astrocyte differentiation. As astrocyte differentiation was affected by TNF treatment, it was logical to investigate the modulation of the 3 main effectors of astrocyte differentiation after TNF induction (Notch, JAK-STAT, BMP pathways). The new title is: NFκB and TNF affect neural stem cell differentiation into astrocytes.
Section 3.1: Figure 1 does not show data concerning changes in differentiation, it rather analyzes apoptosis levels after longer periods of inhibitor treatment, which may be unspecifically toxic to the cells in this case. Alternatively, siRNA experiments can be performed. However, if NFκB is essential for the initiation phase, one would assume that taking away the inhibitor from the cells after 24h of differentiation by a medium change would also block the differentiation but probably enhance cell viability? Furthermore, determining the level of apoptosis by Casp3 gene expression should be replaced by immunoblot analysis of the cleavage product of caspase 3 or of PARP cleavage product. Moreover, in figure 1D also TUNEL staining of later time points of the differentiation process should be shown, as according to the author’s hypothesis after 24, 48 or 72h of cultivation the addition of the inhibitor (for 3 and 6h) does not have an impact on differentiation, therefore one would not expect anymore TUNEL positive cells. This would be important to demonstrate that the impact of NFκB inhibition on survival occurs only at the initiation phase of differentiation, which is a main focus of this work. Finally, TUNEL positive cells should be quantified.
The reviewer is right, Figure 1 does not show data concerning changes in differentiation. However, a treatment with NFkB inhibitor induces a complete cell death in the first 24h of differentiation. It is a drastic process of apoptosis. The proposed siRNA experiments are unfortunately not efficient. To silence NFκB in the cells during the first 24h of differentiation, it would be necessary to transfect them with specific siRNA during the step of proliferation, before the differentiation, thus giving poor gene silencing. Indeed, we have tested this option. This is why we used chemical inhibitors in our study. Specifically, we have used different inhibitors with different mechanisms of action to confirm the observed process of apoptosis (in the article we have provide the results with JSH-23 which present the stronger effect and BAY11-7082 in supplemental data). The reviewer suggests to take away the inhibitor from the cells after 24h of differentiation by a medium change to monitor the differentiation process and cell viability. However, after 24h of differentiation with JSH-23, all the cells were dying as it was shown with the TUNEL analysis (the process of apoptosis is fast and strong). We thank the reviewer for the remark concerning the immunoblot analysis of the cleavage product of caspase 3. We have added this analysis, which shows the cleavage of caspase 3 following a JSH-23 treatment in figure 1C of the revised manuscript. We do not understand the suggestion of the reviewer: “figure 1D also TUNEL staining of later time points of the differentiation process should be shown, as according to the author’s hypothesis after 24, 48 or 72h of cultivation the addition of the inhibitor (for 3 and 6h) does not have an impact on differentiation, therefore one would not expect anymore TUNEL positive cells.” As shown in figure 1B, if JSH-23 is added after the first 24h of differentiation, there are no changes in cell viability in comparison with control conditions. Therefore, in figure 1D the interest of TUNEL staining is to confirm the process of apoptosis observed with JSH-23, when the inhibitor is added to the cells during the first 24h of differentiation. Lastly, the reviewer suggests to quantify TUNEL positive cells. It is important to mention once again that a JSH-23 treatment induces a strong and fast process of apoptosis and this is why we have shown TUNEL staining at 3h and 6h of differentiation. After 24h, there are almost no living cells on the coverslip and after 3h and 6h many cells already detach from the coverslip during the TUNEL protocol. The images showed in figure 1D have been selected where cells were still present in JSH-23 conditions. Thus, this bias prevents us to conduct a proper quantification and statistical analysis.
Section 3.2: As treatment with 50 ng/ml of TNF, which appears to be a very high dose, induces apoptosis in most cell types during the analyzed time points, cell viability should also be checked here as a control.
The reviewer suggests to monitor the cell viability in TNF conditions at different time points along differentiation. In fact, long term TNF treatment induces an increase of apoptotic cells during astrocyte differentiation. However, it is important to note that, together with apoptosis, we observed an increase of the proliferation rate following a TNF treatment. Thus, at the end of the differentiation process, we measured similar number of cells in control or TNF conditions. This was further reflected by similar amounts of RNA levels that we measured between the two conditions. We have added TUNEL assay performed to explore cell death induced by TNF in the revised manuscript in the supplemental data (SD1BC). We have selected this concentration to obtain a strong and continuous activation during the process of differentiation.
Section 3.3: In figure 3A and C do not provide enough evidence concerning the expression level of CD133 or CD44 transcripts over the time of differentiation following TNF-α treatment. For this, one should set the untreated control at time point 0h as 100% and then relate the data to this. This is important because values from untreated samples could also vary, which would have an impact on TNF-α values when the different time points are compared. The same applies for figure 4B. Furthermore, immunocytochemistry is not an adequate tool to quantify protein amounts, but more recommended for localization studies (fig. 3B, D). Figure 4D is more convincing.
Concerning real-time PCR results, we normalized the results to the value 1 for control at each time in each experiment. This methodology was conducted to show the effect of TNF on the relative gene expression during the kinetic analysis of the astrocyte differentiation. Thus, when pooling the different experiments, it appears the value 1 with no error bar in the graph, but for the statistical analysis we compared the delta-CT values obtained in real-time PCR, and of course for the controls, we have detected different CT values (with important variations, which implicate less significance of the results). Concerning the variation of expression during the process of differentiation, we have added in the revised manuscript the kinetic of Cd133 and Cd44 in figure 3AC. The immunocytochemistry pictures in figure 3 are not shown to quantify the protein amounts (WB analysis were unfortunately not convincing (data not shown)), but to highlight the presence and the localization of the protein, which reflects the maintenance of an immature phenotype in TNF treated cells.
Section 3.4: When comparing figure 5A and B one could see that after 24h the TNF-α induced Ki67 transcript values are apperently much lower than after 6h due to the abolished Ki67 expression (Fig. 5A) in untreated samples. Therefore, figure 5B provides a wrong impression and a one-way ANOVA test would probably erase significancies. Data from figure 5B should be integrated in figure 5A. Moreover, authors postulate that there is a correlation between induction of differentiation and inhibition of proliferation. Regarding the effects of TNF-α in this context, can they be reversed by adding the inhibitors? This would enhance the value of the manuscript as they would reveal NFκB-specific influences on proliferation, which are not clear in this section, yet.
As described previously, we normalized the results to the value 1 for control at each time in each experiment. We adopted this methodology to see more clearly the effect of TNF on the relative gene expression during the kinetic analysis of the astrocyte differentiation where we observed important changes in gene expression. In fact, after the first 24h of differentiation, the proliferation was abolished and the reviewer is right that the effect of TNF was small, but this effect was statistically relevant. This observation reinforces the idea that a TNF treatment induces the maintenance of immaturity in these cells and that the differentiation process is delayed in TNF condition. These effects of TNF cannot be reversed by adding JSH23 (data not shown). This is probably due to the fact that RELA/p65 is not implicated in the process of NFκB activation by TNF after the first 24h of astrocyte differentiation.
Reviewer 2 Report
The manuscript, “Astrocyte differentiation is modulated by the NF-kB pathway” submitted by Birck Cindy and coauthors is a valid extension of their previous study. Taken together with their previous report, this study advances the role of NFkB signaling pathways in astrocytes differentiation and de-differentiations. Authors have specifically focused on NFkB P65 subunits, it would be interesting to observe whether P50 subunit of NFkB is also involved in NPC differentiations. I would strongly recommend authors to include experiment on p50 subunit of NFkB and demonstrate whether blocking its activation can modulate NPC – astrocyte differentiations.
The study is planned and discussed well enough. Considering the reproducibility of experimental processes, I would recommend authors to provide catalog numbers of key immunological reagents. Figure 7 can be more informative. Authors should provide the internal funding source for the study in lack of any external sources.
Author Response
The manuscript, “Astrocyte differentiation is modulated by the NF-kB pathway” submitted by Birck Cindy and coauthors is a valid extension of their previous study. Taken together with their previous report, this study advances the role of NFkB signaling pathways in astrocytes differentiation and de-differentiations. Authors have specifically focused on NFkB P65 subunits, it would be interesting to observe whether P50 subunit of NFkB is also involved in NPC differentiations. I would strongly recommend authors to include experiment on p50 subunit of NFkB and demonstrate whether blocking its activation can modulate NPC – astrocyte differentiations.
In our study, we have focused on the modulation of NFκB function during astrocyte differentiation. The mechanistic study concerning the NF-κB family members and their potential effects on cell fate, survival and differentiation goes beyond the scope of the present article (see also the response to reviewer #3). We have shown in supplemental data the localization of NFkB p65 subunit in the nuclei to confirm the activation of NFkB by TNF, but not to investigate the mechanisms underlying these processes.
The study is planned and discussed well enough. Considering the reproducibility of experimental processes, I would recommend authors to provide catalog numbers of key immunological reagents. Figure 7 can be more informative. Authors should provide the internal funding source for the study in lack of any external sources
We thank the reviewer for the positive evaluation regarding our study and the related discussion. For the mentioned suggestions, we have now included the catalog numbers of the immunological reagents in the revised version of the manuscript and added further information to Figure 7 to improve the accuracy of the model (see also the response to reviewer #3). We have now included in Figure 7 the investigated and proposed NF-kappaB subunits relevant for the process of astrocytic differentiation. As mentioned in the manuscript, we have no external sources for the study. The present work was fully supported by the University of Luxembourg, which is now stated in the revised version of the manuscript (MASP, AM and LG supported by ASTROSYS Internal Research Project funding from University of Luxembourg).
Reviewer 3 Report
The manuscript by Birck and coworkers shows a dual regulatory role of NF-kappaB in the differentiation of murine E14 neural progenitor cells into astrocytes: While early inhibition of NF-kappaB prevented differentiation, persistant NF-kappaB-activation was observed to inhibit astrocyte differentiation. Here, TNF-alpha-treatment particularly favored the maintenance of astrocyte precursor profiles in NPCs accompanied by increased proliferation, while Gfap and Aldh1a1 were downregulated. Mechanistically, the authors show persistant NF-kappaB activation to inhibit the astrocytic cell fate via NOTCH signaling (decrease of Hes1 and increase of Hes5 for maintaining glial precursor state) accompanied by down-regulation of the STAT3 and BMP pathway. The present manuscript reports a highly relevant finding with appropriate mechanistic insights regarding a novel crucial dual role of NF-kB in regulating astrocyte differentiation.
Minor comments
- With regard to the distinct roles of NF-kappaB subunits in differentiation of stem cells (e.g. c-REL in fate decision between neurogenesis and oligodendrogenesis, 2020 Apr 22;9(4):1037. doi: 10.3390/cells9041037), the authors should specify the subunits investigated in the abstract to better guide the reader. In addition, the role of distinct NF-κB subunits should be taken into account in more detail in the discussion (currently lines 365-367).
- Graphical overview (Figure 7): Please correct spelling errors. The authors may further include the investigated and proposed NF-kappaB subunits relevant for the process of astrocyte differentiation (see also above).
Author Response
The present manuscript reports a highly relevant finding with appropriate mechanistic insights regarding a novel crucial dual role of NF-kB in regulating astrocyte differentiation.
We thank the reviewer for the very positive feedback regarding the relevance of our studies and the appropriateness of the showed results, which indeed provide critical insights on the role of NF-kB during the astrocytic differentiation.
With regard to the distinct roles of NF-kappaB subunits in differentiation of stem cells (e.g. c-REL in fate decision between neurogenesis and oligodendrogenesis, 2020 Apr 22;9(4):1037. doi: 10.3390/cells9041037), the authors should specify the subunits investigated in the abstract to better guide the reader. In addition, the role of distinct NF-κB subunits should be taken into account in more detail in the discussion (currently lines 365-367).
In our study, we have focused our investigations on the modulation of the NFκB pathway during astrocyte differentiation. Although equally important, a thorough mechanistic study on the NF-κB family members and their potential effects on cell fate, survival and differentiation goes beyond the scope of the present article. As suggested by the reviewer, we have included in the discussion section of the revised manuscript, a paragraph on the distinct NF-κB subunits and their potential roles in astrocyte differentiation.
Graphical overview (Figure 7): Please correct spelling errors. The authors may further include the investigated and proposed NF-kappaB subunits relevant for the process of astrocyte differentiation (see also above).
We thank the reviewer for the remark concerning the spelling errors in Figure 7. They have been corrected in the revised version of the figure. Further, as recommended, we added supplemental elements to Figure 7 in order to improve the accuracy of the model. Indeed, we have now included the investigated and proposed NF-kB subunits relevant for the process of astrocytic differentiation (see also the response to reviewer #2).
Round 2
Reviewer 1 Report
The reply to the comments was appropiate and many points have been adequately addressed. The new title for the manuscript is more appropriate now.However, comments 1 and 2 (from round 1) need to be clarified more (see below). First, to exclude unspecific inhibitor effects and, second, to provide correct information regarding the analyzed cell population.
- To further clarify my suggestion for section 3.1 and figure 1 in the previous review: The authors observed apoptosis within 3-6 hours after starting the cultivation (at 0 hours of differentiation) in the presence of JSH-23 inhibitor (figure 1D). Also the confirmation of these results by demonstrating caspase-3 cleavage product is beneficial for the manuscript (new figure 1C). However, one can not exclude that inhibitor treatment generally leads to apoptosis at later time points of differentiation due to unspecific effects. The MTT assay is not sufficient to exclude apoptosis. Clarifying this would be of importance, here, since the inducibility of apoptosis during the process of differentiation is a read-out for the relevance of NFkappaB. Therefore, data would be more convincing, if TUNEL experiments or caspase-3 cleavage product demonstrate that the differentiated cells (those after 48 or 72 hours of cultivation) do not go into apoptosis when treated for 3-6 h with the inhibitor. This point should be clarified.
- Figures SD1B,C provide the information regarding the respective comment in the previous review. They demonstrate that after treatment with 50 ng/ml TNF there are about 20 to 30% of the cells, which are apoptotic during the differentiation process. This should be mentioned in the main text in section 3.2 because this information is important for the reader when assessing the results of the genome-wide analysis. Conclusively, data reveal not only an inflammatory signature but include at least in part also an apoptotic-related signature.
Author Response
Reviewer#1:
The reply to the comments was appropiate and many points have been adequately addressed. The new title for the manuscript is more appropriate now. However, comments 1 and 2 (from round 1) need to be clarified more (see below). First, to exclude unspecific inhibitor effects and, second, to provide correct information regarding the analyzed cell population.
We thank the reviewer for the positive feedback regarding the revised manuscript.
To further clarify my suggestion for section 3.1 and figure 1 in the previous review: The authors observed apoptosis within 3-6 hours after starting the cultivation (at 0 hours of differentiation) in the presence of JSH-23 inhibitor (figure 1D). Also the confirmation of these results by demonstrating caspase-3 cleavage product is beneficial for the manuscript (new figure 1C). However, one can not exclude that inhibitor treatment generally leads to apoptosis at later time points of differentiation due to unspecific effects. The MTT assay is not sufficient to exclude apoptosis. Clarifying this would be of importance, here, since the inducibility of apoptosis during the process of differentiation is a read-out for the relevance of NFkappaB. Therefore, data would be more convincing, if TUNEL experiments or caspase-3 cleavage product demonstrate that the differentiated cells (those after 48 or 72 hours of cultivation) do not go into apoptosis when treated for 3-6 h with the inhibitor. This point should be clarified.
The reviewer is right, we have not quantified apoptotic cells in the conditions where JSH-23 inhibitor was added to cells after 24h of differentiation, because cell viability was not affected in NSPs treated with JSH-23 after 24, 48, or 72h of differentiation (the visual observation in wells carried out under the phase contrast microscope was confirmed by MTT assay). Thus, the investigations concerning apoptosis where realized only in the conditions in which we observed loss of cell viability (treatment with JSH-23 inhibitor initially during the first 24h of differentiation) to understand the mechanisms implicating in this strong cell death. Concerning the unspecific effects of JSH-23 inhibitor, it’s why we have used an alternative NF-kB inhibitor, BAY11-7082, with a different mode of action, to confirm that NF-kB inhibition induced apoptosis during astrocyte differentiation.
Figures SD1B,C provide the information regarding the respective comment in the previous review. They demonstrate that after treatment with 50 ng/ml TNF there are about 20 to 30% of the cells, which are apoptotic during the differentiation process. This should be mentioned in the main text in section 3.2 because this information is important for the reader when assessing the results of the genome-wide analysis. Conclusively, data reveal not only an inflammatory signature but include at least in part also an apoptotic-related signature.
As suggested by the reviewer, we have now included in the section 3.2 of the revised manuscript, a paragraph concerning the potential role of TNF concerning apoptotic cells in astrocyte differentiation.
Thus, we have added: “As TNF treatment, which appears to induces apoptosis in most cell types, cell viability was be checked here as a control. To study the cell death induced by TNF, we performed a TUNEL assay. To this end, we differentiated NSPs during 6, 24, and 48h with and without TNF. We observed more TUNEL labeled cells in the presence of TNF (SD1B) and cell death was increased at 6, 24, and 48h (30.6%, 21.9%, and 28.0% respectively) (SD1C). However, it is important to note that, together with apoptosis, we observed an increase of the proliferation rate following a TNF treatment. Thus, at the end of the differentiation process, we measured similar number of cells in control or TNF conditions. This was further reflected by similar amounts of RNA levels that we measured between the two conditions (data not shown)”.
Reviewer 2 Report
The manuscript is significantly improved. I would recommend acceptance for the Ms for publication.
Author Response
Response to Reviewer 2 Comments
The manuscript is significantly improved. I would recommend acceptance for the Ms for publication.
We thank the reviewer for the very positive feedback regarding the relevance of our studies and the appropriateness of the showed results, which indeed provide critical insights on the role of NF-kB during the astrocytic differentiation.